# Factors associated with high viral load among HIV clients aged 15 years and older receiving treatment in Tanga city council, Tanzania: A facility-based cross-sectional study

**Aqbara Ibrahim Chande, Novatus A. Tesha**[ORCID]**\*, Bruno Sunguya**

Muhimbili University of Health and Allied Sciences, School of Public Health and Social Sciences 9, United Nations Road, Upanga West, Dar es Salaam, Tanzania

\* teshanovatus@gmail.com

## Abstract

### Background

High viral load indicates poor treatment outcomes among people living with HIV (PLHIV) on antiretroviral therapy (ART). However, there is a dearth of evidence on specific factors associated with high viral load in resource-limited settings, including Tanzania.

### Aim

The aim of this study is to identify factors contributing to high viral load among PLHIV aged 15 years and older on ART for at least six months in Tanga, Tanzania.

### Methods

This is analytical cross-sectional study of 233 PLHIV attending the Care and Treatment Centre (CTC) in Tanga region from September to November 2023. A systematic sampling method was used to select participants for face-to-face interviews. A structured questionnaire was used to collect socio-demographic information while clinical data were collected from the patients' records and CTC database. Descriptive analysis was used to estimate the prevalence of high viral load, while Pearson Chi-square tests compared categorical variables, and the logistic regression assessed determinants of high viral load.

### Results

High viral load was prevalent among 35.2% [95% CI: 29.3%−41.6%] PLHIV attending CTC in Tanga region. Higher viral load was noted among younger adults (52.5%), those in sales/services (63.6%), professionals (54.5%), and unskilled workers (53.2%) compared to their counterparts. PLHIV with severe food

**Data availability statement:** Data pertinent to this study are stored at the Muhimbili University of Health and Allied Sciences' repository. The data is owned by Muhimbili University of Health and Allied Sciences Ethical Clearance Committee which has the legal and ethical mandate to share the data. It may be available to share upon request to the Director of Research and Publications, Muhimbili University of Health and Allied Science, P.0. Box 65001, email: drp@muhas.ac.tz.

**Funding:** The author(s) received no specific funding for this work.

**Competing interests:** This does not alter our adherence to PLOS ONE policies on sharing data and materials.

**Abbreviation:** AIDS, Acquired immune deficiency syndrome; AOR, Adjusted odds ration; ART, Antiretroviral therapy; c-ART, Combined ART; CI, Confidence interval; CTCs, Care and Treatment Centre; HVL, HIV Viral Load; PCA, Principle Component Analysis; PEPFAR, U.S. President's Emergency Plan for AIDS Relief; PLHIV, People living with HIV; SD, Standard deviation; TLD, Tenofovir + Lamivudine + Dolute gravir; UNAIDS, United Nations Programme on HIV/AIDS; VIF, Variance Inflation Factor; VLS, Viral Load Suppression.

insecurity were more likely to exhibit higher viral load compared to those from food secure households (AOR = 9.6; 95% CI: 2.6–35.2). Those consuming alcohol were 4.2 times more likely to have a higher viral load compared to non-drinkers (AOR = 4.2; 95% CI: 1.4–12.5). PLHIV aged 20–24 were 5.4 times more likely to exhibit elevated viral load levels compared to their older counterparts (AOR = 5.4; 95% CI: 1.8–15.7). Additionally, PLHIV in sales and services occupations were 6.1 times more likely to have a higher viral load compared to those in agriculture (AOR = 6.1; 95% CI: 1.1–35 and those facing high stigma were 4.3 times more likely to have a higher HVL than individuals with good social support and low stigma (AOR = 4.3, 95% CI: 1.0–18.7, p = 0.049).).

## Conclusion

More than one in three adult PLHIV on ART in Tanga Tanzania had a high viral load. This burden highlights a steep climb to reach the last 95 target ahead of the deadline. Efforts should focus on young adults, those with households' food insecurity, consuming alcohol, and with perceived stigma in Tanzania and areas with similar context.

## Background

To combat the burden of HIV, the joint United Nations Programme on HIV/AIDS(UNAIDS) launched the "95-95-95" targets in 2015 [1]. The third 95 focuses on viral suppression, where only 53% had reached the agreed target globally [2]. Addressing the non-viral suppression can mitigate the burden of advanced HIV disease, prolong the lives of People Living with HIV (PHLIV), and therefore HIV transmissibility. Advanced HIV disease defined as clinical stages 3 or 4 is marked by severe symptoms like opportunistic infections and significant weight loss, alongside a CD4 count below 200 cells/mm³ [3]. Viral Load Suppression (VLS) is a crucial marker of treatment efficacy in HIV-positive patients and is considered achieved when the reading is less than 1,000 HIV RNA copies/mL of plasma. Despite the effective implementation of *Antiretroviral Therapy (*ART) initiatives in countries with high burden [4], individuals suffering from advanced HIV disease continue to present to hospitals, and as a result, medical admissions and deaths due to HIV-related infectious diseases remain high.

In Tanzania, the adult HIV prevalence rate is 4.4%, with higher rates in women (5.6%) than in men(3.0%) [5]. VLS among adults aged 15 years and older in Mainland Tanzania is 78.1% [5]. According to Tanzania's ART recommendations, successful ART reduces HIV Viral Load (HVL), promotes immunological recovery, and hence increases CD4 cell count [6]. The Tanzania HVL testing guideline, recommends measuring CD4 counts every six months to monitor the immune response to ART [6]. HVL testing is also utilized alongside clinical and immunologic data to diagnose treatment failures more promptly, despite its limited capacity and availability [6,7].

Tanzania remains a priority country for HIV and acquired immune deficiency syndrome (AIDS) in sub-Saharan Africa, with approximately 1.7 million PLHIV as of 2019 year [8]. Njombe region in the Southern highlands of Tanzania reported the highest prevalence at 11.4%, while Tanga in the coastal areas had a moderate rate of 5.0% [8], although higher than the national average. The country has adopted the updated UNAIDS fast track goals to end the AIDS pandemic by 2030, aiming for 95% of PLHIV to know their status, 95% on treatment, and 95% of those on ART to be virally suppressed [9]. By 2022, 83% of HIV-positive individuals were aware of their status, 95% were on ART, and 92% of those on therapy had achieved VLS [9]. These rates of reaching these targets are not similar in all sub-population in the country. For example, the prevalence of VLS among adults aged 15 and older in Mainland Tanzania is only 78.1% [10]. This is below the 95% target for VLS by 2030 aimed at reducing morbidity, mortality, and the spread of HIV [6]. This study aimed to investigate factors influencing HVL among individuals aged 15 years and older receiving HIV treatment in Tanga City Council, Tanzania. This facility-based cross-sectional study aimed to provide insights into the determinants that impact VLS among PLHIV in the region.

## Materials and methods

### Study setting

This analytical cross-sectional study was conducted at four high-volume care and treatment clinics classified as Tier I in Tanga City Council. The facilities serve about 70% of the PLHIV population in the region (Source: City Medical Officer's Office, Tanga). Data were collected from the 4th of September to 27th of November 2023, from Tanga Regional Referral Hospital, Ngamiani, Makorora, and Pongwe Health Centers. The total number of PLHIV across all these facilities was 11,374. These facilities serve half of the region's HIV clients from the 98 of the facilities supported by the U.S. President's Emergency Plan for AIDS Relief (PEPFAR), therefore offering a representative sample for the study. These facilities provided comprehensive HIV care and treatment services free of charge, including HIV testing, treatment, and VL monitoring, accessible to all individuals.

### Study population

This study involved PLHIV aged 15 years and above, who had been on ART for at least six months, attending the four selected facilities. A systematic random sampling was used to recruit eligible participant who consented to participate in the study through written and verbal consent/assent. Patients with serious health complications or mental challenges were excluded from the study.

### Sample size and sampling

To have a representative sample of PLHIV across the four facilities, the study used a sample size formula for finite populations. The total population of PLHIV in the study area was 11,374, with a reported prevalence of unsuppressed HVL of 6.5% in Tanga [10]. For this study, unsuppressed HVL was defined as a plasma HIV RNA level of ≥1000 copies/mL, in line with WHO guidelines [11]. The sample size was calculated using a 95% confidence level (Z = 1.96), a margin of error of 5%($\varepsilon$ = 0.05), and the known population size [12]. The minimum sample size was 73 for each facility. To account for a 5% non-response rate, the final sample size per facility was adjusted to 77, resulting in a total of 308 PLHIV to be recruited across the four facilities. A systematic sampling approach was utilized to select participants, ensuring representation of PLHIV aged 15 years and older on ART for at least six months. Each clinic day, the CTC appointment register was used to identify eligible clients arriving for services. Based on the expected daily attendance and the required sample size, every second eligible client was approached for participation. When a selected client declined participation, or was ineligible, or was not available after the clinical visit, the next eligible client was invited to participate. A total of 243 out of the estimated sample size of 308 participants were recruited, resulting in a response rate of approximately 78.9%. Data of 233(95.9%) participants with HVL tests conducted were therefore analysed further.

## Data collection procedures

Primary and secondary data sources were used in this study. Primary data were collected using a pretested, interviewer-administered Swahili translated questionnaire, adapted from previous studies [13,14]. Secondary data were retrieved retrospectively from patients' records. This included viral load results, history of opportunistic infections, duration on ART, ART regimen, and CTC2 database entries. After being informed about the study objectives, participants who provided written consent were interviewed in Swahili at their Care and Treatment Clinics. Research assistants received training on data collection procedures, participant rights, and rapport building. The questionnaire was pretested at Duga and Tumaini Health Centres to ensure clarity, validity, and reliability. The Content Validity Index (CVI) was 0.86, demonstrating good agreement among experts on the relevance and comprehensiveness of the items. Internal consistency of the multi-item scales was also confirmed, with Cronbach's alpha values ranging from 0.78 to 0.85, indicating acceptable reliability.

## Study variables

The dependent variable was HVL, defined as a viral load ≥1000 copies/mL based on routine viral load monitoring records documented in the CTC2 database and facility laboratory reports. HVL was categorized as "yes" (coded as 1) for participants with viral load ≥1000 copies/mL and "no" (coded as 0) for those with viral load <1000 copies/mL, independent variables included socio-economic and demographic factors such as age, gender, education, occupation, marital status, smoking status, alcohol consumption [14], stigma (measured through the Abridge Scale) [15], household food insecurity, socio-economic factors (wealth index generated through principle component analysis – PCA)[14]; and clinical data (WHO HIV clinical stage, c-ART type, duration, and presence of opportunistic infections such as Tuberculosis, Cryptococcal infection, and toxoplasmosis. Stigma was measured using the Abridged HIV Stigma Scale, a validated multi-item instrument assessing perceived HIV-related stigma. Responses were scored and summed to generate a total stigma score, which was then categorized into low and high stigma based on predefined thresholds. Food insecurity was assessed using the Household Food Insecurity Access Scale (HFIAS).

## Data analysis

This study employed both descriptive and inferential analyses. Descriptive statistics summarized data using frequencies/counts and mean ± standard deviation (SD) for continuous variables, while percentages were used for categorical variables. The proportion of HVL (>1000 copies/mL) was assessed at a 95% confidence interval and tabulated against participants' sociodemographic and clinical characteristics using a Pearson Chi-square test. For inferential analyses, multivariate logistic regression estimated the association between independent variables and HVL. Initially, univariate analysis explored unadjusted associations. In the adjusted logistic regression model, a forward stepwise method was used to fit the predictors of HVL where one independent variable was added into a model at a time. The variables which showed multicollinearity (Variance Inflation Factor (VIF)>4) and those with p > 0.2 in the unadjusted model were excluded. Probability > 0.2 were excluded to avoid overfitting the model with variables which might not be significant [16]. However, important sociodemographic and clinical characteristics variables of interest were retained regardless of statistically insignificant in the unadjusted model except those with multicollinearity. Statistical significance was set at p < 0.05. Data analysis was conducted using Stata/MP 14.2 (StataCorp LLC).

## Ethical considerations

Ethical clearance was granted by Muhimbili University of Health and Allied Sciences (DA.282/298/01.C/1878). Permission was obtained from the City Medical Officer and healthcare officials. Written and verbal consent was obtained from participants with an assent for those below 18 years of age. Participants were informed of their rights, including withdrawal at any time without affecting care they receive. Unique identity numbers were used and no personal identifiers were

collected. Throughout data collection and analysis we maintained confidentiality. Data was stored on a password-protected computer.

## Results

### Sociodemographic and clinical characteristics of study participants

Majority of participants were aged above 24 years 172(73.8%), the mean age was 32.4 years (±11.4 SD), with females making up 144(61.8%). 134(68.4%) had pre and primary education. Majority of participant were on Combined ART (c-ART) of Tenofovir + Lamivudine + Dolutegravir (TLD) regimen 220 (94.4%) with no participant on HIV clinical stage 4. High HVL was noted among 82(35.2%) [95% CI; 29.3%−41.6%] of the participants with HVL tests results available (Table 1).

### Viral load levels by sociodemographic characteristics of the study participants

High HVL prevalence was noted among 32(52.5%) young adults, 14(63.6%) participants from sales and services industry, 6(54.5%) professionals, and 25(53.2%) unskilled labor categories. Additionally, high HVL was prevalent among 59(52.2%) of participants who consume alcohol and 70(39.5%) participants with high level of stigma. Moreover, high HVL was prevalent among 33(82.5%) participants with severe household food insecurity (**Table 2**).

### Viral load by clinical characteristics of the study participants

Participants on the TLD c-ART regimen had a lower proportion of high HVL 73(33.2%) compared to those on other regimens 9(69.2%, p = 0.008). Those on c-ART for 13–24 months had a higher proportion of high HVL rate of 29(74.4%) than those on c-ART for over 25 months (26.5%, p = 0.001). Proper storage of c-ART significantly reduced high HVL 60(29.0%, p = 0.001). Additionally, individuals in WHO HIV clinical stage three had a higher HVL proportion of 10(76.9%) compared to those in stage one 50(33.6%) and two 22(31.0%, p = 0.005). **Table 3**

### Predictors of high viral load among study participants

Several variables met the inclusion criteria for the regression model, having p-values less than 0.2, indicating their potential association with high viral load. These variables included age groups (p = 0.001), alcohol consumption (p = 0.001), stigma status (p = 0.015), food insecurity severity (p = 0.001) and c-ART storage in the container (p = 0.001).

Young adults aged 20–24 were 5.4 times more likely to have a higher HVL than those aged 25 and older (AOR = 5.4, 95% CI: 1.8–15.7, p = 0.002). PLHIV working in sales and services were 6.1 times more likely to have a higher HVL compared to those in agriculture, (AOR = 6.1, 95% CI: 1.1–35.4, p = 0.044). Alcohol consumption increased the likelihood of a higher HVL by 4.2 times compared to non-drinkers (AOR = 4.2, 95% CI: 1.4–12.5, p = 0.011), while those facing high stigma were 4.3 times more likely to have a higher HVL than individuals with good social support and low stigma (AOR = 4.3, 95% CI: 1.0–18.7, p = 0.049). PLHIV experiencing severe food insecurity were 9.6 times more likely to have a higher HVL than those with mild to moderate food insecurity (AOR = 9.6, 95% CI: 2.6–35.2, p = 0.001). Furthermore, improper storage of c-ART outside its original container increased the risk of a higher HVL by 6.7 times compared to those who properly stored their medication (AOR = 6.7, 95% CI: 1.2–38.1, p = 0.033). (**Table 4**)

## Discussion

This study found a high HVL prevalence of 35.2% [95% CI; 29.3%−41.6%] among PLHIV aged 15 and older on c-ART for at least 6 months. Higher proportions were found in young adults, varied with occupations, individuals experiencing stigma, those from severe food insecurity households, from households with low wealth index, traveling long distances to health facilities, those on non-TLD regimens, those on c-ART for 12–24 months, and with advanced WHO HIV stages.

**Table 1. Sociodemographic and clinical characteristics of study participants.**

| Characteristics | | Frequency | Percentage |
|---|---|---|---|
| **Age groups in years** | Young adults 15–24 | 61 | 26.2 |
| | Adults >24 | 172 | 73.8 |
| **Mean age in years** | 32.4 (SD: 11.4) | | |
| **Gender** | Male | 89 | 38.2 |
| | Female | 144 | 61.8 |
| **Education level** | Pre & Primary | 134 | 68.4 |
| | Secondary O-Level | 48 | 24.5 |
| | Secondary A-level and above | 14 | 7.1 |
| **Occupation** | Professional/technical/managerial | 11 | 4.7 |
| | Clerical | 8 | 3.4 |
| | Sales and services | 22 | 9.4 |
| | Skilled manual | 27 | 11.6 |
| | Unskilled manual | 47 | 20.2 |
| | Domestic service | 46 | 19.7 |
| | Agriculture | 72 | 30.9 |
| **Marital status** | Never married | 32 | 13.7 |
| | Married | 112 | 48.1 |
| | Widowed | 21 | 9.0 |
| | Divorced | 11 | 4.7 |
| | Separated | 32 | 13.7 |
| | Cohabiting | 25 | 10.7 |
| **Smoking status** | Never smoked | 195 | 83.7 |
| | Currently smoking | 38 | 16.3 |
| **Alcohol consumption** | Yes | 113 | 48.5 |
| | No | 120 | 51.5 |
| **Stigma status** | Low | 56 | 24.0 |
| | High | 177 | 76.0 |
| **Food insecurity** | Yes | 77 | 33.0 |
| | No | 156 | 67.0 |
| **Food insecurity severity** | Mild to moderate | 113 | 73.9 |
| | Severe | 40 | 26.1 |
| **Household wealth** | Low | 131 | 56.2 |
| | Medium | 67 | 28.8 |
| | High | 35 | 15.0 |
| **Is distance a big problem** | Yes | 19 | 8.2 |
| | No | 214 | 91.8 |
| **Clinical characteristics** | | | |
| **c-ART regimen** | TLD | 220 | 94.4 |
| | Others | 13 | 5.6 |
| **Duration on c-ART** | ≤ 12 months | 13 | 5.6 |
| | 13-24 months | 39 | 16.7 |
| | ≥ 25 months | 181 | 77.7 |
| **c-ART stored in a container** | Yes | 207 | 88.8 |
| | No | 26 | 11.2 |
| **WHO HIV Clinical Stage** | Stage 1 | 149 | 63.9 |
| | Stage 2 | 71 | 30.5 |
| | Stage 3 | 13 | 5.6 |
| | Stage 4 | 0 | 0.0 |

*(Continued)*

**Table 1.** (Continued)

| Characteristics | | Frequency | Percentage |
|---|---|---|---|
| **Opportunistic infection** | Yes | 48 | 20.6 |
| | No | 185 | 79.4 |
| **Viral load** | Low | 151 | 64.8 |
| | High | 82 | 35.2 |

**Table 2. Viral load by sociodemographic among PLHIV (n = 233).**

| Variable | | Viral load | | P-value |
|---|---|---|---|---|
| | | N (%) | N (%) | |
| | | Low 151 (64.8) | High 82 (35.2) | |
| **Mean age in years** | 32.4 (SD: 11.4) | | | |
| **Age groups in years** | Young adults 15-24 | 29 (47.5) | 32 (52.5) | **<0.001*** |
| | Adults >24 | 122 (70.9) | 50 (29.1) | |
| **Gender** | Male | 55 (62.1) | 34 (37.9) | 0.478 |
| | Female | 96 (66.7) | 48 (33.3) | |
| **Education level(n=196)** | Pre & Primary | 82 (61.2) | 52 (38.8) | 0.335 |
| | Secondary O-Level | 31 (64.6) | 17 (35.4) | |
| | Secondary A-level and above | 6 (42.8) | 8 (57.2) | |
| **Occupation** | Professional/technical/managerial | 5 (45.5) | 6 (54.5) | **<0.001*** |
| | Clerical | 7 (87.5) | 1 (12.5) | |
| | Sales and services | 8 (36.4) | 14 (63.6) | |
| | Skilled manual | 17 (63.0) | 10 (37.0) | |
| | Unskilled manual | 22 (46.8) | 25 (53.2) | |
| | Domestic service | 32 (69.6) | 14 (30.4) | |
| | Agriculture | 60 (83.3) | 12 (16.7) | |
| **Marital status** | Never married | 21 (65.6) | 11 (34.4) | 0.318 |
| | Married | 77 (68.8) | 35 (31.2) | |
| | Widowed | 12 (57.1) | 9 (42.9) | |
| | Divorced | 9 (81.8) | 2 (18.2) | |
| | Separated | 20 (62.5) | 12 (37.5) | |
| | Cohabiting | 12 (48.0) | 13 (52.0) | |
| **Smoking status** | Never smoked | 130 (66.7) | 65 (33.3) | 0.178 |
| | Currently smoking | 21 (55.3) | 17 (44.7) | |
| **Alcohol consumption** | Yes | 54 (47.8) | 59 (52.2) | **<0.001*** |
| | No | 97 (80.8) | 23 (19.2) | |
| **Stigma status** | Low | 44 (78.6) | 12 (21.4) | |
| | High | 107 (60.5) | 70 (39.5) | **0.013*** |
| **Food insecurity** | Yes | 32 (41.6) | 45 (58.4) | **<0.001*** |
| | No | 119 (76.3) | 37 (23.7) | |
| **Food insecurity severity(n=153)** | Mild to moderate | 71 (62.8) | 42 (37.2) | **<0.001*** |
| | Severe | 7 (17.5) | 33 (82.5) | |
| **Household wealth** | Low | 75 (57.2) | 56 (42.8) | **<0.001*** |
| | Medium | 42 (62.7) | 25 (37.3) | |
| | High | 34 (97.1) | 1 (2.9) | |
| **Distance is a big problem** | Yes | 3 (15.8) | 16 (84.2) | **<0.001*** |
| | No | 148 (69.2) | 30.8) | |

**Note:** Asterisks (*) indicate statistically significant values (P<0.05).

**Table 3. Viral load by clinical factors among PLHIV (n = 233).**

| Variables | | HIV Viral load | | P-Value |
|---|---|---|---|---|
| | | Low-N (%) | High-N (%) | |
| c-ART regimen | TLD | 147 (66.8) | 73 (33.2) | 0.008* |
| | Others | 4 (30.8) | 9 (69.2) | |
| Duration on c-ART | ≤ 12 months | 8 (61.5) | 5 (38.5) | <0.001* |
| | 13-24 months | 10 (25.6) | 29 (74.4) | |
| | ≥ 25 months | 133 (73.5) | 48 (26.5) | |
| c-ART stored in a container | Yes | 147 (71.0) | 60 (29.0) | <0.001* |
| | No | 4 (15.4) | 22 (84.6) | |
| WHO HIV Clinical Stage | Stage 1 | 99 (66.4) | 50 (33.6) | 0.005* |
| | Stage 2 | 49 (69.0) | 22 (31.0) | |
| | Stage 3 | 3 (23.1) | 10 (76.9) | |
| Opportunistic infection | Yes | 31 (64.6) | 17 (35.4) | 0.971 |
| | No | 120 (64.9) | 65 (35.1) | |

**Note:** Asterisks (*) indicate statistically significant values (P < 0.05).

Similar studies in Rwanda and Cameroon reported a prevalence of 29% and 20.6% [13,17], particularly among young adults and those on c-ART for 12–24 months.

This study found that severe food insecurity is associated with elevated VL among PLHIV. Alongside c-ART, a balanced diet is essential for improving health and nutritional status. These findings align with previous research linking food insecurity to higher VL, highlighting the importance of ensuring food security and access to essential nutrients for vulnerable populations [18]. Hunger can lead to medication discontinuation and reduced care attendance, negatively affecting adherence and contributing to psychological disorders.

Participants with high stigma levels in this study were more likely to have high HVL, likely due to poor communication with healthcare providers and reduced treatment adherence. Stigma may also worsen mental health issues like depression [19], contributing to treatment fatigue. A study in Dar es Salaam and other contexts found similar effects of stigma on VLS [20,21],. suggesting that addressing stigma and psychological issues could reduce HVL and prevent treatment failure.

Alcohol consumption, especially binge drinking, can impair judgment regarding medication adherence and increase engagement in high-risk sexual behaviours, leading to poor adherence and new HIV infections [22], which may result in persistently high HVL. This study's findings align with research from Rwanda, showing significantly higher odds of high HVL among alcohol consumers [13]. Similarly, previous studies have found a strong association between alcohol use and increased odds of high HVL among PLHIV on c-ART [23]. Providing intensive counselling among PLHIV on alcohol cessation or moderation can help reduce HVL and the risk of treatment failure.

This study found that young adults had significantly higher odds of high viral loads compared to older participants, likely due to their relatively lower capacity to mobilize resources for ongoing care. Factors such as cognitive and physical development, mental health, substance use, and family support can impact treatment adherence among adolescents and young adults [13]. The Tanzania HIV Impact Survey (2022/23) indicated that younger adults were less likely to achieve VLS than their adult counterparts. These findings align with studies in Botswana [24] and the U.S. [25], which also showed that younger adults struggle more with treatment adherence and achieving viral load suppression compared to older adults.

Studies shows occupation significantly impacts medication adherence and VLS among PLHIV [7,26]. The study found that participants working in sales, skilled manual, and unskilled manual occupations were more likely to have elevated

**Table 4. Factors Associated with Higher viral load among PLHIV aged ≥15 years.**

| Characteristics | Viral load | High n % | Univariate | | Multivariate | |
|---|---|---|---|---|---|---|
| | Low n% | | COR(95% CI) | P value | AOR (95% CI) | P value |
| **Age groups in years** | | | | | | |
| Young adults 15–24 | 29 (47.5) | 32 (52.5) | Reference | | | |
| Adults >24 | 122 (70.9) | 50 (29.1) | 2.8 [1.6-5.1] | **0.001*** | 5.4 [1.8-15.7] | **0.002*** |
| **Sex** | | | | | | |
| Female | 96 (66.7) | 48 (33.3) | Reference | | | |
| Male | 55 (62.1) | 34 (37.9) | 1.2 [0.7-2.1] | 0.478 | 1.2 [0.4-3.4] | 0.77 |
| **Occupation** | | | | | | |
| Agriculture | 60 (83.3) | 12 (16.7) | Reference | | | |
| Professional/technical/managerial | 5 (45.5) | 6 (54.5) | 5.5 [1.5-20.9] | **0.012*** | 7.5 [0.9-60.7] | 0.059 |
| Clerical | 7 (87.5) | 1 (12.5) | 0.7 [0.1-5.8] | 0.708 | 1.3 [0.05-33.3] | 0.888 |
| Sales and services | 8 (36.4) | 14 (63.6) | 8.7 [3.0-24.6] | **0.001*** | 6.1 [1.1-35.4] | **0.044*** |
| Skilled manual | 17 (63.0) | 10 (37.0) | 3.5 [1.4-9.0] | **0.008*** | 8.6 [1.7-43.8] | **0.010*** |
| Unskilled manual | 22 (46.8) | 25 (53.2) | 5.7 [2.5-12.9] | **0.001*** | 8.1 [2.0-32.7] | **0.003*** |
| Domestic service | 32 (69.6) | 14 (30.4) | 2.6 [1.1-5.7] | **0.036*** | 3.3 [0.7-16.3] | 0.138 |
| **Smoking** | | | | | | |
| Never smoked | 130 (66.7) | 65 (33.3) | Reference | | | |
| Currently smoking | 21 (55.3) | 17 (44.7) | 1.6 [0.8-3.3] | 0.181 | 0.5 [0.1-2.3] | 0.378 |
| **Alcohol consumption** | | | | | | |
| No | 97 (80.8) | 23 (19.2) | Reference | | | |
| Yes | 54 (47.8) | 59 (52.2) | 4.6 [2.6-8.3] | **0.001*** | 4.2 [1.4-12.5] | **0.011*** |
| **Stigma** | | | | | | |
| Low | 44 (78.6) | 12 (21.4) | Reference | | | |
| High | 107 (60.5) | 70 (39.5) | 4.6 [2.6-8.3] | **0.015*** | 4.3 [1.0-18.7] | **0.049*** |
| **Food insecurity severity** | | | | | | |
| Mild to moderate | 71 (62.8) | 42 (37.2) | Reference | | | |
| Severe | 7 (17.5) | 33 (82.5) | 8.0 [3.2-19.6] | **0.001*** | 9.6 [2.6-35.2] | **0.001*** |
| **Clinical factors** | | | | | | |
| **c-ART stored in the container** | | | | | | |
| Yes | 147 (71.0) | 60 (29.0) | Reference | | | |
| No | 4 (15.4) | 22 (84.6) | 11.6[3.8-34.8] | **0.001*** | 6.7 [1.2-38.1] | **0.033*** |

Note: COR – crudes odds ratio, AOR – Adjusted odds ratio

Asterisks (*) indicate statistically significant values (P<0.05, OR>1)

HVL. This may be due to tight work schedules and concerns about job security, which can limit their ability to attend clinic appointments and adhere consistently to ART.

Studies have reported conflicting results whereby a study in Ghana did not found a link between employment and HVL [27], while a study in Nigeria suggested unemployment correlates with better VLS [28]. Furthermore, a meta-analysis study indicated that employed individuals generally adhere better to treatment [29]. These conflicting results highlight the complexity of the relationship between employment status and VLS among PLHIV. Nonetheless, it is essential to consider occupation as a potential influencing factor when initiating c-ART for people living with HIV.

## Strength and limitation

This study has several strengths, including its focus on PLHIV in Tanga, Tanzania, the use of probability sampling, and standardized data collection tools, which enhance the relevance and reliability of the findings. The study also provides important insights into the influence of stigma, food insecurity, and substance use on viral load suppression among PLHIV. However, several limitations should be acknowledged. The cross-sectional design limits the ability to establish causal relationships between variables. The study relied on self-reported information, which may be subject to recall and social desirability bias. Viral load measurements were based on a single time point, which may not fully capture longitudinal treatment outcomes. The response rate was 78.9%, with non-participation due to absent clients, ineligibility, or refusal, which may introduce selection bias. Some adjusted odds ratios had wide confidence intervals, likely due to small cell counts or limited sample sizes, reducing estimate precision. Finally, the measurement of economic status may not have fully captured participants' socioeconomic conditions.

## Conclusion and recommendation

The high prevalence of elevated viral load among PLHIV on c-ART for six months is a concern, with 1 in 3 at risk of treatment failure. Key predictors include sociodemographic factors, food insecurity, stigma, alcohol use, and employment. Targeted interventions like food availability support and stigma reduction are vital.

## Author contributions

**Conceptualization:** Aqbara Ibrahim Chande, Novatus A. Tesha, Bruno Sunguya.

**Supervision:** Aqbara Ibrahim Chande, Novatus A. Tesha, Bruno Sunguya.

**Writing – review & editing:** Aqbara Ibrahim Chande, Novatus A. Tesha, Bruno Sunguya.

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
