## [Decision Letter · Decision Letter 0]

21 Jan 2026

PONE-D-25-20662Factors associated with high viral load among HIV clients aged 15 years and older receiving treatment in Tanga city council, Tanzania: A facility-based cross-sectional studyPLOS One

Dear Dr. Tesha,

Thank you for submitting your manuscript to PLOS ONE. After careful consideration, we feel that it has merit but does not fully meet PLOS ONE’s publication criteria as it currently stands. Therefore, we invite you to submit a revised version of the manuscript that addresses the points raised during the review process.

The manuscript addresses an important public health issue: determinants of high viral load among PLHIV in Tanzania:using primary data from care and treatment centres. The topic fits well within PLOS ONE’s scope of public health, epidemiology, and determinants of health outcomes. The study is relevant for programmatic decision‑making, especially in resource‑limited settings.

However, there are several methodological, reporting, and structural weaknesses that must be addressed before the manuscript meets PLOS ONE’s publication standards.

**Major Concerns**

**1. Study design and analysis inconsistencies**

The manuscript states inclusion of adults **≥15 years**, yet regression analysis uses a “20–24” category, skipping 15–19 years—unclear if 15–19 were excluded or combined elsewhere.Reporting of logistic regression is inconsistent: variables retained “regardless of significance” conflict with the stepwise exclusion logic described.

**2. Sampling and sample size issues**

Initial sample size was **308**, but only **243** were enrolled (78.9% response). Reasons for non‑participation and implications for bias are not discussed.Systematic sampling approach is insufficiently described (sampling interval, starting point, handling of absent clients).

**3. Outcome definition inconsistency**

VL >1000 copies is defined as high viral load, but tables sometimes use “greater than 1000 copies/dl”—a nonstandard unit that must be corrected.

**4. Potential misclassification of variables**

Time on ART categories (≤12 months, 13–24, ≥25 months) are used, but it is unclear whether VL tests were first, routine, or repeat measures.Food insecurity and stigma scales need clearer description of scoring, thresholds, and validation in this population.

**5. Interpretation appears overstated**

Some adjusted ORs have extremely wide confidence intervals (e.g., 1.1–35.4; 1.2–38.1), suggesting imprecise estimates due to small cell counts.The discussion occasionally implies causation, which is inappropriate for a cross-sectional design.

**6. Writing, structure, and clarity**

Several grammatical errors, missing words, and formatting issues require revision.The background section mixes national statistics with global information in a way that feels unfocused.

**Minor Concerns**

Some references are outdated or inconsistent in formatting.Tables are lengthy and may benefit from consolidation or improved formatting.Ethical section appears duplicated and should be edited for clarity.

We look forward to receiving your revised manuscript.

Kind regards,

Hamufare Dumisani Mugauri, Ph.D. Medicine and Health Sciences

Academic Editor

PLOS One

Journal Requirements:

The authors declare that they have no known competing financial interests or personal relationships that could have appeared to influence the work reported in this paper.

Reviewers' comments:

Reviewer's Responses to Questions

**Comments to the Author**

1. Is the manuscript technically sound, and do the data support the conclusions?

Reviewer #1: Yes

Reviewer #2: Yes

2. Has the statistical analysis been performed appropriately and rigorously? 

Reviewer #1: I Don't Know

Reviewer #2: Yes

3. Have the authors made all data underlying the findings in their manuscript fully available?

Reviewer #1: Yes

Reviewer #2: Yes

4. Is the manuscript presented in an intelligible fashion and written in standard English?

Reviewer #1: Yes

Reviewer #2: Yes

5. Review Comments to the Author

Reviewer #1: The Editor

PLOS ONE Journal

13 November 2025

RE: Reviewer Comments to the Author on

Factors associated with high viral load among HIV clients aged 15 years and older receiving treatment in Tanga city council, Tanzania: A facility-based cross-sectional

(Manuscript ID: PONE-D-25-20662)

Dear PLOS ONE Editor,

I would like to thank the authors for their contribution to this important topic. The manuscript addresses factors associated with high viral load among HIV clients aged 15 years and older receiving treatment in Tanga City Council, Tanzania, using a facility-based cross-sectional design. The study is relevant and provides valuable insights for HIV management and public health strategies in the region.

Below, I provide detailed reviewer comments to improve the clarity, rigor, and overall quality of the manuscript.

Reviewer Comments

1. Abstract

Please add a heading titled “Aim” in the manuscript to clearly state the main objective of the study.

2. Materials and Methods: Study Setting

The manuscript states that the facilities serve about 70% of the PLHIV population in the region. Please provide a reference to support this statement.

3. Study Population / Inclusion Criteria

Please provide more details on how study participants were recruited. If convenient, clarify the informed consent process, specifically, whether consent was obtained verbally or in writing.

4. Sample Size and Sampling

4.1 The manuscript mentions a total population of 11,374 PLHIV and a 6.5% prevalence of unsuppressed high viral load (HVL) in Tanga. Please kindly define the threshold or level of viral load used to categorize it as “unsuppressed.

4.2 Sample Size Inconsistency

The manuscript states that a total of 308 PLHIV were to be recruited across the four facilities, but the Methods section and the Abstract subheading mention 243 participants. Please clarify this discrepancy to ensure consistency.

5. Data Sources, Collection, and Tools

The manuscript states that the English questionnaire was translated into Swahili for clarity and pretested at non-research sites (Duga and Tumaini Health Centres) to ensure validity and reliability. Could the authors clarify whether any multi-item scales were included in the questionnaire? If so, please provide the Cronbach’s alpha values or other reliability statistics to demonstrate internal consistency.

6. Data Analysis

6.1 The manuscript defines high viral load (HVL) as a viral load greater than 1000 copies/dL. Please clarify the units used, as the standard reporting for HIV viral load is in copies/mL, not copies/dL

6.2 . The manuscript states that a stepwise method was used in the multivariable logistic regression to identify predictors of high viral load. Please clarify which stepwise approach was applied (forward, backward, or both), and specify the criteria for variable entry and removal (e.g., p-value thresholds or information criteria) to enhance transparency and reproducibility of the analysis.

6.3 Please mention the full term Variance Inflation Factor (VIF) before using the abbreviation. Also, clarify the rationale for excluding variables with p > 0.2 in the unadjusted model.

7. Tables 1, 2, and 3

It is recommended to add a footnote indicating that p ≤0.05 was considered statistically significant for the variables in the tables. This will help clarify the interpretation of significant results for readers.

7.1 Table 2

The results describe high viral load proportions for WHO clinical stages two and three, but stage four is not mentioned. Please clarify whether data for stage four were unavailable, excluded, or not applicable, and if available.

7.2 Table 2

Please define the abbreviations ‘HAART’ and ‘TLD’ at their first occurrence in a footnote, to ensure clarity and consistency for readers unfamiliar with these terms.

7.3 Table 3 (Regression Model)

Please clarify the criteria used to select variables included in the regression model for Table 3, titled “Factors Associated with Higher Viral Load among PLHIV aged ≥15 years.” Specify whether inclusion was based on statistical significance in univariate analysis, theoretical relevance, or other selection criteria.

7.4 Table 3

Please remove the repeated word “Note” in the footnote under Table 3.

8. Headings Please add the following headings to the manuscript to meet journal requirements and enhance clarity and completeness:

• Authors’ Contributions

• List of Abbreviations (in alphabetical order)

Summary and Recommendation

Thank you for considering my revisions.

This manuscript, “Factors Associated with High Viral Load among HIV Clients Aged 15 Years and Older Receiving Treatment in Tanga City Council, Tanzania: A Facility-Based Cross-Sectional Study,” addresses an important and timely topic with clear relevance to HIV program improvement and public health practice. However, minor revisions are required to meet academic and journal standards fully.

The suggested improvements will enhance clarity, methodological transparency, and overall presentation.

Recommendation: Minor Revision

Best regards,

Zainab Mahmood Al-Zadjali

Reviewer, Department of Family Medicine and Public Health, College of Medicine and Health Sciences, Sultan Qaboos University, Oman

Reviewer #2: Factors associated with high viral load among HIV clients aged 15 years and older receiving treatment in Tanga city council, Tanzania: A facility-based cross-sectional study

Thank you for the opportunity to review the manuscript titled: Factors associated with high viral load among HIV clients aged 15 years and older receiving treatment in Tanga city council, Tanzania: A facility-based cross-sectional study

Some minor comments.

Abstract

This part is well written and contains all the elements of a good abstract, the following are minor correction to make:

1. In the background part of the abstract: Line 4 add the word “years” after “15”

2. Method part of abstract: Recheck and clarify the correct long form of CTC as authors have written it as care and testing center but from what I know is care and treatment centre (or clinic). Also, the author needs to add a statement specifying that the clinical data were collected from patient’s record to be clear to the reader.

3. Conclusion part of abstract: The author recommended about stigma, but this was not reported anywhere in the result section.

Background:

This is also well written, but minor corrections to make in the following parts

4. Paragraph 2 Line 2: Add the word “years” after “15”

5. Paragraph 2 Line 3: What is the long form of VL? (Recommended all abbreviation to be written in full in their first mention with abbreviation in brackets, then continue with abbreviation in the subsequent use of the term)

6. Paragraph 2 Line 4: Which practice is being referred here? Please specify.

7. Paragraph 3 Line 2: Add the word “year” before “2019”, and the authors are recommended to use updated reference for example there are data for year 2022, check and correct word spacing throughout the document.

8. Paragraph 3 Line 2: What is the long form of HVL? (Recommended all abbreviation to be written in full in their first mention with abbreviation in brackets, then continue with abbreviation in the subsequent use of the term).

Methods:

By and large this section is okay, except for minor comments under the following parts:

Study setting

9. Line 3: The author needs to correct the word “of” which is written in superscript back to normal

10. Line 7: The sentence “all facilities provided comprehensive HIV…………………………….” Is not clear to the reader, please rephrase.

Study population

11. Line 1: suggested the first sentence to begin as “This study involved PLHIV aged 15 years and above, who had……………………..”

Data sources, collection and Tools

12. Suggested the subheading (Data sources, collection and Tools) to be changed and written as “Data collection procedure”

13. Line 1: Change the words “in the study” to be “in this study”

14. Line 2: Add words “Swahili translated” before the word “questionnaire”

15. Line 3: Replace the word “time” with the word “duration”

16. Line 5: Suggested, the sentence “After identifying participants……” to be rephrased and written as “identified participants were interviewed using Swahili translated questionnaire………”

17. Line 8: The sentence “The English questionnaire was translated………………” is not needed here as the information has already been said above, so suggested to delete it.

Study variables

18. Recommended all abbreviation to be written in full in their first mention with abbreviation in brackets, then continue with abbreviation in the subsequent use of the term

19. The word “HAART” sounds ancient terminology nowadays the word combined antiretroviral therapy (c-ART) is used, so suggested to replace the word “HAART” with “c-ART” throughout the document

Data analysis

20. The author needs to describe the test used to compare proportions (low versus high viral load) for the results presented in table 1 and 2

21. Check and correct word spacing

Ethical considerations

22. The subheading to be re-written as “Ethical considerations” (with an “s” added on word “consideration”)

23. Lin 3: For the participants under 18 years normally seek for assent and not consent. Please re-check and correct accordingly.

24. Line 4: Did withdrawal from the study affected their cate at the CTC? if not suggested to add a statement “without affecting care they receive” after the word “time”

Results

This is well written, but the following are minor comments for improvement:

25. The author is advised to add another table as table 1 with the table title “Sociodemographic and clinical characteristics of study participants” which will have 3 columns as exemplified below:

Characteristics Frequency (n) Percentage (%)

This will give a general picture of characteristics of participants involved in this study.

Viral load by sociodemographic characteristics of the respondents

26. Suggested the subheading to be re-written as “Viral load levels by sociodemographic characteristics of the study participants”

27. I don’t think logistic regression analysis was used to determine the significance of difference in proportion of viral load level between variables. The author needs to re-check and clarifies this.

Viral load by clinical characteristics of the study participants

28. Check word spacing and correct

29. Indicate the p-value in brackets for all variables that showed significant difference

Predictors of high viral load among study participants

30. In statement form, first write the independent variables related to your outcome variables on Univariate. Then, on Multivariate, write the factors related to your outcome variable (higher viral load) by interpreting the association using AOR and CI (as described currently)

31. Table 3: Amend the table column heading as suggested below:

Characteristics Viral load Univariate Multivariate

Low n (%) High n (%) COR (95% CI) P value AOR (95% CI) P value

This will make the reader to clearly follow and understand

32. Make OR under the univariate "COR" and multivariate "AOR" in table 3, and write the whole form as footnotes beneath the tables

33. There is no p value with asterisk (*) in the table 3 rather bolded p value. If you use asterisk (*) , this will better be presented as footnote beneath the table in the following format: “

*Statistically significant

Discussion

34. Paragraph 1 line 1: Present prevalence with CI in brackets

35. Paragraph 1 line 4: replace “late WHO HIV stage” with “advanced HIV stages”

36. Paragraph 6 line 1: suggested to rephrase the sentence “Literature shows mixed findings……………………………..” re-writing as “Studies have reported conflicting results whereby a study in Ghana did not find a link between employment and HVL while a study in Nigeria suggested unemployment correlates with better VLS. Furthermore, a meta-analysis study indicated that employed individuals generally adhere better to treatment.”

Conclusion and recommendation

37. Line 1: Change the words “is concerning” to be “is a concern”

38. Recommendations and conclusions should be grounded in the findings of your study, particularly when recommending any accountable body, which should be based on factors related to your outcome variable (high viral load).

Grammar and missing/repeated words

39. Please check over grammar, spelling, word spacing and missing/duplicated words throughout the document and correct.

40. The manuscript is not numbered line-by-line in order to provide simple peer-review, please add this for re-submission

6. PLOS authors have the option to publish the peer review history of their article (what does this mean?). If published, this will include your full peer review and any attached files.

Reviewer #1: **Yes:** Zainab Mahmood Dilshad Al Zadjali

Reviewer #2: **Yes:** Shabani Iddi

<scribe-shadow data-crx="ejommccbnocgekjphflienmbmkallcef" id="crxjs-ext" style="position: fixed; width: 0px; height: 0px; top: 0px; left: 0px; z-index: 2147483647; overflow: visible; visibility: visible;"></scribe-shadow>

---

## [Author Response · Author response to Decision Letter 1]

13 Apr 2026

Thank you, all reviewers' comments have been addressed and are attached as a point-to-point matrix. We thank the PLOS ONE team and reviewers for their comments, which have further shaped our work.

---

## [Editor Report · Decision Letter 1]

23 Apr 2026

Factors associated with high viral load among HIV clients aged 15 years and older receiving treatment in Tanga city council, Tanzania: A facility-based cross-sectional study

PONE-D-25-20662R1

Dear Dr. Tesha,

We’re pleased to inform you that your manuscript has been judged scientifically suitable for publication and will be formally accepted for publication once it meets all outstanding technical requirements.

Kind regards,

Hamufare Dumisani Mugauri, Ph.D. Epidemiology and Public Health

Academic Editor

PLOS One
---

## [Editor Report · Acceptance letter]

PONE-D-25-20662R1

PLOS One

Dear Dr. Tesha,

I'm pleased to inform you that your manuscript has been deemed suitable for publication in PLOS One. Congratulations! Your manuscript is now being handed over to our production team.

Kind regards,

on behalf of

Dr Hamufare Dumisani Mugauri

Academic Editor

PLOS One